# Situational Analysis of Cat Ownership and Cat Caring Behaviors in a Community with High Shelter Admissions of Cats

**DOI:** 10.3390/ani14192849

**Published:** 2024-10-03

**Authors:** Jacquie Rand, Rebekah Scotney, Ann Enright, Andrea Hayward, Pauleen Bennett, John Morton

**Affiliations:** 1Australian Pet Welfare Foundation, Kenmore, QLD 4069, Australiadolly5664@gmail.com (A.H.); 2The University of Queensland, Faculty of Science, School of Veterinary Science, Gatton Campus, Brisbane, QLD 4343, Australiarebekah.scotney@uq.edu.au (R.S.); johnmorton.jemora@gmail.com (J.M.); 3School of Psychology and Public Health, La Trobe University, Bendigo, VIC 3552, Australia; pauleen.bennett@latrobe.edu.au; 4Jemora Pty Ltd., P.O. Box 5010, East Geelong, VIC 3219, Australia

**Keywords:** feline, cat ownership, disadvantaged community, sterilization, cat management, cat containment, One Welfare

## Abstract

**Simple Summary:**

Management of stray cats in urban areas is an ongoing challenge in Australia, and many are euthanized, particularly in disadvantaged regions. The Australian Pet Welfare Foundation aimed to assess the impact of a free cat sterilization program in Ipswich, Queensland. Before that intervention, a situational analysis was conducted to evaluate cat and dog ownership behaviours. In a sample of 343 residents from the area, 35% owned cats, and 3% fed strays. Cats were mainly obtained from family or friends and shelters, while 53% of respondents owned dogs, mostly sourced from breeders and family acquaintances. A higher percentage of owned cats (91%; but only 74% for those aged 4 to <12 months) were sterilized compared to owned dogs (78%). Among cat owners, 51% contained their cats all the time and 18% at night. Our findings suggest that community-based sterilization programs targeting both owned and semi-owned cats, and assisting semi-owners in becoming owners, would assist in reducing unplanned litters and the stray cat population. It is also recommended that assistance with cat containment be provided where vulnerable native species are present in urban and peri-urban areas. These insights are crucial for developing effective policies aligned with One Welfare principles.

**Abstract:**

Managing stray cats in urban areas is an ongoing challenge, and in Australia, many are euthanized. Most stray cats are from disadvantaged areas and are under 1 year of age. The Australian Pet Welfare Foundation intended to assess the impact of a free cat sterilization program in an area with high shelter intake of cats in the city of Ipswich, Queensland. The aims of this pre-intervention study were to undertake a situational analysis of cat ownership, semi-ownership and cat caring behaviours, and compare those in the same demographic with dog ownership and caring behaviours relating to sterilization rates, to provide a basis against which to assess the program’s effectiveness. In a sample of 343 participants from that area, 35% owned cats and 3% fed stray cats. Cats were predominantly obtained from family or friends (31%) and shelters (20%). More respondents owned dogs (53%), which were most often sourced from breeders (36%) and family acquaintances (24%). More owned cats than owned dogs were sterilized (91% versus 78%). However, only 74% of cats aged 4 to <12 months were sterilized. Cat containment practices varied, with 51% of owners containing their cat(s) at all times, and a further 18% doing so at night. These results suggest the need for community-based programs that focus on sterilizing owned and semi-owned cats, and assisting semi-owners in becoming full owners to reduce stray cat populations and associated negative impacts. This includes assistance with cat containment where vulnerable native species are present. Public understanding of the causes and effective solutions for free-roaming cats, alongside legislative changes, are required to facilitate these efforts. Assistive programs aligned with One Welfare principles are expected to benefit the wellbeing of animals, humans and their environments.

## 1. Introduction

In most countries, management of stray cats in urban and peri-urban areas is a longstanding and continuing challenge [1,2]. There are an estimated 0.7 million urban stray cats in Australia (plausible range: 0.07–2.56 million) [3] and, based on the human population at the time (2016) [4] this represents 29 cats per 1000 residents (range: 3–104 cats/1000 residents). Stray cats in urban and peri-urban areas of Australia are categorized by the Royal Society for the Prevention of Cruelty to Animals (RSPCA) as domestic cats [5]. Domestic cats are those with some dependence (direct or indirect) on humans and are subcategorized into owned, semi-owned and unowned cats [5].

Currently, in Australia, there is an estimated population of 5.3 million owned cats [6]. Semi-owned cats, also considered stray cats, receive intentional care such as feeding from people who do not perceive themselves as their owners [7,8,9]. Approximately 3–10% of Australian adults feed an average of 1.5 cats daily that they do not perceive they own [8,9]. These cats vary in sociability, with many socialized to humans, and may be associated with one or more households. Unowned cats, also considered stray cats, receive food from humans unintentionally, such as from food waste bins, and are indirectly dependent on humans [5].

In contrast to the US, Australia defines feral cats as those living remotely from humans. They are considered a pest species and can legally be poisoned, shot or killed with blunt trauma [5,6,10,11]. They are distinguished from domestic cats because they have no relationship with or dependence on humans, survive by hunting or scavenging, and live and reproduce in the wild [5]. Because feral cats do not live where people live or frequent, they are not a source of complaints regarding nuisance behaviours. In contrast, in the US and many other countries, this distinction is not made, and the term “feral” is also used to describe cats that are less socialized or unaccustomed to humans, but live around where humans live or frequent [12,13]. They are often fed by well-meaning people, and hence in Australia, would be considered semi-owned domestic cats, and those receiving food unintentionally from humans would be considered unowned domestic cats [5].

The majority of cats entering animal welfare shelters and council pounds (municipal animal facilities) are classed as strays, originate from low socioeconomic areas, and were born in the preceding 6 to 12 months [14,15]. Although feral cats, by definition, do not enter shelters and pounds in Australia, stray cats are sometimes misclassified as feral based on behaviour shortly after admission, leading to euthanasia despite evidence showing this is an invalid way to distinguish feral cats [16,17]. In Australia, stray cats represent 85–100% of cats admitted to pounds and 60–80% of cats entering animal welfare shelters, with the remainder being surrendered by owners [14,15]. Because most cats entering shelters and pounds lack microchips or collars [18], it is difficult to distinguish between owned, semi-owned and unowned cats. However, recent Australian research suggests that most urban stray cats are likely to be unidentified owned or semi-owned cats, with unowned cats being uncommon [19,20].

Many stray cats and kittens admitted into animal shelters and municipal pounds are euthanized despite being healthy or treatable [2,14], negatively impacting the mental health of staff involved and causing community discord. Shelter staff are often required to regularly kill healthy and treatable cats and kittens, increasing their risk of depression, traumatic stress, substance abuse, high blood pressure, sleeplessness and suicide [21,22,23,24,25,26,27,28]. On average across Australia, approximately 33% of cats entering shelters and pounds are euthanized, with an estimated 50,000 euthanized annually in Australia [2]. The majority are healthy stray cats. In comparison, an estimated 1.8 million cats are euthanized annually in the US (17% of cat admissions) in animal welfare shelters and municipal facilities [29]. In the UK, the reported euthanasia rate is approximately 13% of cats admitted to animal welfare shelters and rescue groups [30].

Typical approaches to manage urban and peri-urban free-roaming cats in Australia include reactive ‘trap, adopt or kill’ methods and mandated cat containment, both of which have proven to be ineffective at reducing the number of free-roaming cats or alleviating associated issues in the medium and long-term [2,31,32,33,34]. Current Australian data suggest that approximately 7% [2,3] of the urban stray cat population is killed in shelters and council pounds annually. This low-level ad hoc culling is insufficient to reduce the overall number of stray cats over time, due to the high cat reproductive rate, immigration of new cats into the area, and increased survival of juveniles [35,36]. High-level culling or sterilization is required to sustainably reduce urban stray cat populations [37]. However, high-level culling is cost prohibitive and socially unacceptable [8], with no published reports of successful implementation at a suburb or city level. This method of domestic cat management is also not aligned with One Welfare principles, which seek to optimize the wellbeing of animals, humans and their physical and social environments [1,38,39].

Despite many Australian councils (local governments) implementing by-laws mandating cat containment, these are not an effective strategy for reducing free-roaming cats and, in the medium and long-term, have been found to increase cat-related complaints, increase trapping and impoundment of cats, increase council costs, result in higher numbers of cats being killed [2,20], and subsequently increase exposure of staff to negative impacts on job satisfaction and mental health [31,34]. The failure of mandated containment as an effective strategy is principally because most free-roaming cats in urban areas are not owned and therefore do not have an owner to contain them. Others are unidentified owned cats, which hinders the effectiveness of enforcement action directed at the owner. For cats with owners, property limitations and financial barriers, as well as concerns about cat welfare, can be barriers for containment [40,41].

Modern sheltering practices are increasingly focused on proactive strategies to reduce numbers of cats admitted to shelters and thus numbers euthanized [42]. These include high-intensity sterilizing programs targeted to areas of high shelter intake [20,43,44]. In Australia, legal restrictions under biosecurity, abandonment, and containment laws make trap-neuter-return (TNR) programs for semi-owned and unowned cats illegal [1,45]. Instead, Australian sterilization programs focus on owned cats and assisting semi-owners to take ownership of the cat/s they are caring for [20]. International and Australian scientific evidence repeatedly demonstrates that sterilization programs that are targeted and have sufficient intensity will significantly reduce the number of cats impounded and euthanized by municipal authorities and shelters, and significantly reduce cat-related calls to local government authorities [43,44,46,47,48,49]. Recent modelling data from the UK determined that the pet cat population was the biggest contributor to the stray cat population, and that sterilization rates of 95% and 98% were required to stabilize or decrease the stray cat population, respectively [50].

The Australian Pet Welfare Foundation intended to implement a targeted, high-intensity sterilization program, offering free sterilization, microchipping, vaccination, and parasite treatment for all owned and stray domestic cats in the targeted areas, chosen because they had high cat intake into the local shelters. The aim was to investigate the effects on shelter intake and euthanasia of cats, as well as the number of free-roaming cats and cat-related complaints. The ‘Community Cat Program’ was covered by a research permit approved by the Queensland Government under a Department of Agriculture and Fisheries Scientific Research Permit (No. PRID000825) for handling of “restricted matter”, which includes all cats that are not owned. This allowed cats remaining semi-owned or unowned to be legally sterilized, and the authors believe this is the first such permit which has been issued in Australia.

Prior to institution of a Community Cat Program, we conducted a situational analysis of cat ownership, semi-ownership and cat caring behaviours and compared those in the same demographic with dog ownership and caring behaviours relating to sterilization rates and unplanned breeding. The study was implemented to provide a basis for a subsequent impact analysis on cat semi-ownership, sterilization and unplanned litters as a result of the program. The purpose of this paper is to report findings from this situational analysis.

## 2. Materials and Methods

Data were obtained from residents in three suburbs within the city of Ipswich, Queensland (Goodna, Rosewood, Redbank Plains), with a total population of 38,003 [51,52,53]. These suburbs were identified as having the highest per capita shelter and impound admissions in Ipswich (>20 cats per 1000 residents) and, therefore, were selected to trial a Community Cat Program based on free sterilization and microchipping of owned and stray cats. The Socio-Economic Indexes for Areas (SEIFA) scores for socioeconomic advantage and disadvantage in 2021 for the three suburbs (855, 884 and 934 for Goodna, Redbank Plains and Rosewood respectively) were less than the average for the Ipswich local government area (940) and for Australia overall (1003), where a lower score indicates greater social disadvantage [54]. These suburbs had higher proportions of residents in rental housing than the average for Australia (Australia 31%; Redbank Plains 58% [51]; Goodna 48% [52]; Rosewood 41% [53], and median personal weekly incomes were lower than for Australia collectively (Australia AUD 805; Redbank Plains AUD 747 [51]; Goodna AUD 637 [52]; and Rosewood AUD 636) [53]. Rosewood is a small rural town of approximately 3000 residents, whereas Goodna and Redbank Plains are two adjacent suburbs closer to the central business district of the city of Ipswich. The Queensland Government Animal Management (Cats and Dogs) Act 2008 requires microchipping of cats by 12 weeks of age [55] and owner details lodged on a recognized database within 7 days of implantation. The Ipswich City Council Local Law No. 6 (Animal Management) requires all owned cats to be contained to the owner’s property and a permit obtained for keeping 3 or more cats. [56]. Neither the State nor the city of Ipswich have laws or by-laws requiring mandatory sterilization or additional registration (licensing).

A survey was undertaken using online questionnaire software provided by ArcGIS Survey123 (Redlands, CA, USA) [57]. Respondents were requested to provide information on their gender, suburb, age, education, culture they most identified with, pet ownership, their interactions with different types of animals and their beliefs about stray cat management (Table A1). Only results relating to cat and dog ownership, semi-ownership and caring behaviours are presented in this paper.

The questionnaire (Table A1) was administered from 6 June 2020 to 25 September 2021, initially face-to-face (31 respondents) and then by telephone (330 respondents) due to restrictions associated with the COVID-19 pandemic. Respondents to the door-knock or telephone call who were 18 years or older and an Australian resident, were invited to participate anonymously in the survey. Whoever answered the door or phone was asked to complete the survey if eligible. During the consent process, potential respondents were informed that the “purpose of this survey is to gauge community attitudes towards pet ownership and the management of unowned and stray cats and kittens living in the city and suburbs”.

The door-knock routes used were those previously chosen for walking transects for counting cats, and along those routes any house where the front door was accessible, and it appeared safe to enter the property, was approached. Of the 31 residences recorded as being approached, responses from one person from each of 28 residences were obtained. For the remaining 3 residences, no respondent was enrolled for the following reasons: householder declined, person available was aged under 18 years so was ineligible, or no one was home (1 residence each).

Landline and mobile telephone numbers for the 3 suburbs were obtained from Sample Pages (Cremorne, Victoria, Australia) for 6004 phone numbers (2252 landline; 3752 mobile) which were filtered to remove people on the do not call list. The list provided “may also have contained numbers included in the list by Sample Pages to ensure that the numbers were being used in accordance with the agreement” (which specified duration they could be used for). Of these 6004 phone numbers, 3190 phone numbers were randomly chosen to be called, of which 2169 (68%) were called once, 766 (24%) were called twice or more and 255 (8%) were called three times or more. Of those, 1116 (35%) calls went straight to voice mail and 383 (12%) were not answered, resulting in 1499 (47%) people not being reached for communication. Of the remaining 1691 (53%) potential participants who answered the call, 240 (14.2%) were no longer living in the area, 71 (4.2%) of calls ended immediately, 3 (0.2%) were busy but happy to reschedule but no one got back to them and 1047 (61.9%) declined further communication; resulting in 330 (19.5%) participants who answered the phone and who agreed to participate in the survey, and of these, 315 eligible responses were provided.

Of the 3190 phone numbers called, 1380 (1691-240-71) answered the phone and were known to live in the area, and of these, 330 residents agreed to participate and 315 provided an eligible response via phone (response rate of those who answered the phone and lived in the area 315/1380 = 22.8%). However, the response rate for all phone numbers called (3190), was only 11.5%, if 14.2% of these 3190 numbers were to people who no longer lived in the area. Thus, it is assumed we called 3190 × (100–14.2)/100, and used data from 315 respondents enrolled by phone, equating to a response rate for phone call enrolments of 315/2737 or 11.5%.

Thus, data were entered for a total of 361 residences (31 in person, 330 by phone) and for 343 of these (28 and 315, respectively), an eligible respondent provided responses. Of those who specified a residential postcode (n = 320), 2.5% (8) gave a postcode not associated with the three targeted suburbs. All 8 were retained in the study because, prior to participating in the survey, it was confirmed that the person did reside in the nominated area. Hence, it was assumed the postcode was recorded incorrectly when the questionnaire was administered. Of the 343 respondents, the questionnaire was completed in full by 96% (330/343; 78% (22/28) of the face-to-face and 98% (308/315) of the telephone interviews), demonstrating a good level of engagement. The numbers of eligible responses obtained from the target suburbs were in approximate proportion to the suburbs’ human population sizes.

Respondents were asked about whether they owned any pets and, if so, select details about their cat(s) and dog(s) (Table A1). Respondents were asked for their degree of agreement with statements “I like cats”, “I like dogs” and “I like native animals” and degree of agreement was assessed by a five-point Likert scale (strongly disagree, disagree, neutral, agree, strongly agree). Respondents were also asked whether they fostered wildlife, cats or dogs and if they fed or cared for animals they did not own, such as cats, dogs, birds, possums or other animals. Denominators for reported results vary due to non-responses to some questions and inconsistent responses to different questions by individual respondents. In addition, some analyses are for specific subsets of the 343 respondents. Ninety-five percent confidence intervals for proportions were calculated using the -proportion- command in Stata (version 18, StataCorp, College Station, TX, USA). Logit-transformed confidence intervals were calculated. For proportions of respondents’ cats or dogs, robust (Hubner-White or sandwich) standard errors that accounted for clustering of cat or dog within respondent were used. The study was approved by The University of Queensland Human Ethics Committee (approval number 2014000597).

## 3. Results

Among the respondents, 34% were male and 66% were female, with the median age bracket being 35 to 39 years. The majority (87%) identified most with Australian culture (Table A2). In terms of education, 49% had completed only primary or secondary school, 27% held a certificate diploma, and 24% held a graduate or postgraduate university degree. A total of 339 respondents answered the question, “Do you own a pet?” Of these, 75% (254) owned a pet. Specifically, 35% owned at least one cat (some may have also owned other animals), 53% owned at least one dog, 9% owned one or more birds, 6% owned fish, 3% owned chickens, 1% owned at least one horse, and 1% owned at least one reptile.

### 3.1. Cat Ownership

Of the 332 respondents who recorded their gender, 38% (85/221) of female respondents were cat owners, compared to 29% (32/111) of male respondents. Of the 116 cat owners reporting the number of cats they owned, most had one cat (51%), but 36% had two cats and 13% had 3 or 4 cats. The greatest proportion of these cats had been obtained from family or friends (31%), followed by from a shelter or council pound (20%), and 19% were found (Table 1). The most common age categories of cats were adults (50%, aged 1–6 years) and seniors (34%, aged 7–15 years; Table 2). The majority of cats were sterilized at the time of completing the questionnaire (91%, or 180/197; 95% CI: 86% to 95%), although only 74% (14/19; 95% CI: 46% to 90%) of those aged 4 to 11 months were sterilized, compared to 96% (164/171; 95% CI: 92% to 98%) for those ≥1 year of age.

Of the cat owners, 7% (8/115) reported that at least one of their cats had a litter of kittens after coming into their care. For 2 of these 8 respondents, at least one of their cats had a litter in the last twelve months. For all 8 respondents, the litters were accidental (i.e., unplanned), with the main reason for the litters being that the cat got out accidentally and got pregnant (5 of the 8 respondents). Another was the result of a stray cat arriving while pregnant, with 3 respondents (including one who selected ‘got out accidentally’) stating that they did not get around to desexing the cat. Kittens from the 8 accidental litters were given away or sold on-line (3 respondents); given or sold to friends or family (2 respondents); or surrendered to the council, kept one and gave rest to friends, or not yet decided (1 respondent each). For 5 of these 8 owners, all female cats owned at the time they completed the questionnaire were now sterilized (1 cat for 4 respondents and 2 cats for 1 respondent). However, for another 2 of the 8, their single female cat was not sterilized at that time, while the final respondent in this group had one female cat but was unsure about its sterilization status.

### 3.2. Dog Ownership

Of the 332 respondents who recorded their gender, 57% (126/221) of female respondents were dog owners, compared to 45% (50/111) of male respondents. Of the 168 dog owners reporting the number of dogs they owned, most had one dog (61%), but 36% had two dogs and 4 respondents had more (3, 4, 6 and 10 dogs, respectively). Of the 258 owned dogs whose source was described by respondents, 36% obtained their dogs from a breeder (either registered or unregistered), followed by 24% who obtained their dog from a family friend (Table 1).

The most common age categories for dogs were adults (52%, 1–6 yr) and seniors (38%, 7–15 yr; Table 2). Most (73%; 190/260; 95% CI: 66% to 79%) were sterilized at the time of completing the questionnaire, although of dogs aged 4 to 11 months, only 44% (8/18; 95% CI: 22% to 69%) were sterilized, compared to 76% (182/238; 95% CI: 69% to 83%) of dogs ≥1 yr.

Of the dog owners, 4% (7/163) reported that at least one of their dogs had a litter of puppies after coming into their care. For 3 of these 7 respondents, at least one of their dogs had a litter in the last 12 months. Of the 7 respondents whose dog had puppies after coming into their care, 3 of the litters were planned and 4 were accidental (i.e., unplanned). Reasons for the accidental litters included: the dog got out accidentally and got pregnant (2 of the 4), and the bitch became pregnant to her son (1 of the 4). The fourth respondent stated they were ‘not really concerned about having puppies’. Puppies from the 7 respondents were given away or sold on-line (2 respondents), given or sold to friends or family (4 respondents), or kept at least one and gave or sold the rest to friends or family (1 respondent). Of these 7 respondents, 6 owned female dogs at the time the questionnaire was administered. Two of these 6 respondents reported that their single female dog was sterilized but all female dogs owned by the remaining 4 respondents were not sterilized at that time (1 female dog for 3 respondents, and 3 female dogs for 1 respondent).

### 3.3. Cat Containment

Of the 119 cat owners, 117 described their cat(s) containment, and nearly all (85% or 99/117; 95% CI: 77% to 90%) contained their cat(s) to their property always or for some of the time. Approximately half (51%) reported that they always contained their cats, a further 33% confined their cat/s some of the time (including 18% (21/117) who contained them always at night), and 15% did not contain their cat(s) (Table 3).

Of the 60 respondents who reported always containing their cat to their property, 68% kept their cat indoors entirely, with the remainder containing them to their property, mostly with some indoor access. However, 3 of these 60 respondents reported that at least one of their cats had escaped or wandered off their property in the previous 2 weeks.

### 3.4. Feeding Unowned Animals and Preferences for Various Types of Animals

Of respondents, 5% (17/329) fostered animals for a wildlife service or shelter, and 5% (18/330) fostered dogs and/or cats from their local shelter or rescue group, including 3% (10/328) who did both. In addition, many respondents regularly fed and/or cared for animals that they did not own, nor were fostering or minding: 24% (82/343) fed birds, 9% fed cats, 8% fed dogs, 4% fed possums, and 2% fed other species, including kangaroos, lizards, horses and frogs. Gender was not specified for 7 respondents. Among the remaining 223 females (F) and 113 males (M), however, there were both similarities and differences in the percentages who fed and/or cared for: birds (F 24%, M 24%); cats (F 11%, M 4%); dogs (F 7%, M 9%); possums (F 4%, M 3%); and other species (F 3%, M 2%).

Of the 30 people feeding and/or caring for cats they did not own, 26 provided information about these cats. Of these 26, 42% (11) indicated they fed at least one cat that they did not know to be owned by someone else. Since between 0 and 4 of the 4 respondents who failed to provide this information would have done likewise, we can conclude that between 3% and 4% (11 to 15/343) of respondents were cat semi-owners, caring for a stray cat. These cat semi-owners were mostly female (82%, 9/11), ranged in age from 22 to 81 years (median 56 years), and 73% (8/11) also owned pet cats. People feeding dogs they did not own were not asked about the ownership status of those dogs.

Respondents’ degrees of agreement with statements “I like dogs very much”, “I like cats very much”, and “I like native birds and animals very much” were assessed using 5-point scales. Proportions of respondents strongly agreeing were 76% (256/335) for dogs, 41% (136/332) for cats and 73% (240/330) for native birds and animals. In comparison, only 2% strongly disagreed that they liked dogs, compared to 18% who strongly disagreed that they liked cats. By gender, 75% (166/221) of females and 79% (88/112) of males strongly agreed to liking dogs, 46% (101/219) of females and 31% (34/111) of males strongly agreed to liking cats, and 72% (158/219) of females and 74% (814/109) of males strongly agreed to liking native birds and animals.

Not surprisingly, more cat owners strongly agreed to liking cats (73% of cat owners) than dog owners (37%) or bird owners (37%) (Table A3). More bird owners liked dogs (73%) than cats (37%). Similar proportions of dog and cat owners strongly agreed to liking native birds and animals (79% and 76%, respectively), but a greater proportion of bird owners liked native birds and animals (86%).

## 4. Discussion

In Australia, management strategies for free-roaming urban cats remain ineffective, with animal control focusing on the symptoms and typically implementing very basic de-population methods [1,2]. Despite minor changes, these decades-old cat management programs are still commonly used to control urban stray and peri-urban cat populations [1].

They have not achieved sustained reductions in cat-related complaints, or the number of cats impounded and euthanized [2,58], and have negatively affected the job satisfaction and mental health of many shelter and municipal staff. Cat population management programs based on sterilizing semi-owned and unowned cats (TNR) in target areas are increasingly shown to be effective in reducing shelter intakes but are illegal in Australia. However, in a first for Australia, a Department of Agriculture and Fisheries research permit for “restricted matter” was obtained to allow sterilization of semi-owned and unowned stray cats. Prior to implementing a targeted, high-intensity sterilization program for owned, semi-owned and unowned cats, a situational analysis was conducted to assess its effectiveness in increasing sterilization rates and cat caring behaviours. Comparisons were made with select dog caring behaviours related to sterilization and unplanned litters.

### 4.1. Demographics

Although women constitute 51% of the population in Queensland and nationally [59], women constituted 66% of our respondents. This is consistent with findings in the international literature, whereby women are generally more likely to be enrolled in surveys than men, especially phone-based surveys. Studies suggest that women may be more accessible during survey times and potentially more willing to participate in survey research [60,61]. Gender imbalances were also identified in previous studies about cats, where 78% to 85% of respondents identified as female [9,62,63]. This may in part be attributed to women respondents being more involved with caring for pet and stray cats [9,62,63]. This was supported by our finding that a greater proportion of women than men identified as owning cats and dogs.

The median age bracket of respondents was 35 to 39 years of age, which is aligned with Queensland’s median age of 38 years [64]. Consistent with the lower SEIFA scores for the suburbs selected for our study (defined as lower access to material and social resources, and less ability to participate in society), education levels were lower than the general Australian population. For example, 48% of the Australian population have non-school qualifications below a university degree, while 31% hold a university bachelor’s degree or higher [59] compared to only 27% and 24%, respectively, in our cohort. Lower socioeconomic levels are often associated with reduced access to higher education opportunities [59]. Information regarding income was not collected in this study but would be useful to include in future studies, as would housing type.

### 4.2. Pet Ownership

Across Australia, the proportion of residents owning pets is reported to be 69% [65], which is lower than the 75% reported by our respondents, but very similar to the 76% reported from the adjacent city of Brisbane (population 2.3 million), where respondents were selected from a representative distribution of socioeconomic backgrounds [8]. Dog ownership was more common than cat ownership in our cohort (53% versus 35%). Surprisingly, the Brisbane study found that slightly more respondents owned a cat than a dog (56% versus 52%), although more females answered the survey, likely resulting in skewed results [8]. In addition, a greater proportion of residents live in a flat or apartment in Brisbane compared to Ipswich (26% [66] versus 2% [67]), which are often more suited to cats than dogs.

The Australian cat ownership rate is 33% [68], but in some European countries, cat ownership is substantially higher. For example, 48% of residents in Russia, 41% in Poland and 37% in Latvia own at least one cat [69]. This could be attributed to cultural factors, urban living conditions, and differing pet ownership traditions [70,71]. In 2022, 72% of the population living in European cities lived in a flat or apartment [71] where cats, being low-maintenance and adaptable to smaller living spaces, are preferred over dogs. In the European Union (EU), the most common companion animal species are cats [72] Notably, in our study, only 41% of respondents liked cats, compared to 76% liking dogs. This is possibly associated with frequent media releases in Australia portraying cats, including pet cats, as “natural born killers of wildlife” [73] and “decimating native wildlife” [74]. This messaging is intentionally aimed at obtaining a social license to manage feral cats with lethal means, and pet cats by mandated containment [6].

#### 4.2.1. Sterilization Rates and Accidental Litters

At the time of completing the survey, 91% of cat-owners reported that all of their owned cats were sterilized (83% of cats were aged ≥ 1 year), which was unexpectedly high given the demographics and is similar to the 2019 Brisbane study (93% sterilized) [8]. In 2024, 89% of cats in Australia were sterilized compared to 81% of dogs [75].

Our sterilization rates are also higher than those reported from national surveys from the UK (85%) [76] and Germany (80%) [77]. This might be attributed to more accessible and subsidized sterilization programs. For example, the National Desexing Network, an initiative of the Animal Welfare League, Queensland, provides highly subsidized sterilization programs across Australia through participating councils and veterinary practices, with both contributing a subsidy [78].

Despite generally high sterilization rates, only 74% of owned cats aged 4 to 11 months in our study were sterilized, consistent with lower rates being reported in younger cats elsewhere. Of cats presenting to a free microchipping clinic in Western Australia, only 49% of cats under 2 years of age were sterilized, compared to 93% of older cats [79]. This highlights opportunities for free and affordable sterilization to be targeted to newly acquired cats and kittens in areas with high cat and kitten intake into shelters.

In contrast to cats, only 73% of dogs were sterilized. This may reflect veterinary advice to delay sterilizing large breed dogs until older than 12 months to reduce the risk of orthopaedic issues and certain cancers associated with early sterilization. These include increased likelihood of hemangiosarcoma, lymphoma, mast cell tumours, and canine cruciate ligament (CCL) rupture in neutered dogs [80,81]. However, most dogs in our study were aged over 12 months and only 74% of those aged 1–6 years and 79% of those aged 7 to 15 years were sterilized. Alternatively, or in addition, dogs may be perceived as easier to contain than cats, with reduced opportunities for unplanned pregnancies, lowering the perceived importance of sterilization compared to cats [82]. Dog sterilization surgeries are also generally more costly than for cats.

Cat sterilization may also reflect a desire to control unwanted behaviours, such as noisy mating calls, particularly of females, and fighting, roaming and odorous spraying, often seen in intact male cats, which are significantly reduced following neutering [83].

More cat owners (7%) than dog owners (4%) reported that their pet had a litter after coming into their care and, for all cat owners this was accidental, whereas nearly half the dog owners reported that the litter was planned. Research indicates that many cat owners are unaware that cats can reach sexual maturity as early as four months old, which can lead to unplanned pregnancies if the cats are not sterilized in time [84,85]. In addition, many veterinarians in Australia still recommend 6 months as the ideal age for sterilization [86,87].

Mandating sterilization has not been successful in reducing overpopulation of cats in Australia, because the three states with mandated sterilization have the highest per capita shelter intakes and numbers of cats euthanized compared to all other states not mandating it [2]. Household income is the strongest predictor of whether a cat is sterilized [88,89]. For low-income households, the priority and urgency are for funding food, rent, electricity and transport and there is less priority or urgency for funding sterilization of cats, especially those passively acquired. The consequences of an accidental litter are substantially less than not being able to fund essential household needs. Unwanted kittens are either given away, surrendered to a shelter, pound or veterinary clinic, or allowed to wander. Although 75% of free-living kittens die [90] before 6 months of age, those that survive may find someone else to care for them, as 19% of cat owners did for cats in our study.

To reduce unwanted kittens being born, resources should be directed to lower socio-economic areas to improve timely sterilization of cats because these communities have lower rates of early sterilization and higher incidences of unplanned litters [79]. This needs to be coupled with information on feline reproductive capabilities to prevent early pregnancies, and multiple accidental litters [91,92]. A shift by veterinarians towards offering early-age sterilization of cats in Australia is strongly recommended, moving from the traditional six months to before four months. This is safe, reduces surgery time and prevents early pregnancies, thus decreasing unwanted litters [91].

Based on modelling, the pet cat population in the UK was the biggest contributor to the stray cat population, and it was calculated that sterilization rates of 95% and 98% were required to stabilize or decrease the stray cat population, respectively [50]. This assumed that pet cats constituted 92% of the urban cat population. In Australia, based on the estimate in 2016 of 3.3 million pet cats and 0.7 million stray cats [3], pet cats would constitute 79% of all cats, with stray cats (semi-owned and unowned) being approximately 21%. The stray cat population in Australia does not appear to be decreasing, based on static or increasing numbers of cats recorded as being impounded by councils [2]. Therefore, the sterilization rate in our study of 91% across all ages, and 74% for cats 4–11 months of age, indicates that earlier and higher sterilization rates are required in the owned cat population to positively impact the stray cat population. However, because of the greater proportion of semi-owned and unowned cats in Australia, and likely other countries such as the US, sterilization programs also need to target these populations, especially because most are not sterilized. Legislation urgently needs to be amended in Australia to facilitate sterilization of semi-owned and unowned cats. This is especially important for management of multi-cat sites.

#### 4.2.2. Sources of Acquisition

The sources of cat acquisition were different from dogs, with the largest proportion of respondents acquiring their cat from a family member or friend, whereas the most common source for dog owners was from a breeder. Passive acquisition occurs when cats are unintentionally obtained, including being found or obtained through social networks or received as gifts from family or friends [7]. This includes acquisition when someone in their network is no longer able to care for their cats due to life changes, (e.g., entering nursing home, passing away, imprisonment, relocating overseas or extended travel) [93]. There was no difference in attachment between owners with pets who were, or were not, acquired intentionally, and one study reported that pets acquired unintentionally as gifts or free were less likely to be relinquished than pets acquired intentionally [93,94]. Recent surveys report approximately 40–48% of cat owners passively acquire their cat for free, especially in lower socioeconomic areas [7,95], which is consistent with our study, where 54% of cats came from family or friends or were found. Higher populations of stray or unowned cats are reported from lower socio-economic areas [20,38] and these provide a free, ready source of cats for local residents. Cats obtained from shelters, pounds and rescue or foster care groups often come with basic veterinary care already provided, such as sterilization, vaccination, parasite treatment and microchipping, which is included in the adoption price. There are also opportunities for discussions with new owners about the cats’ needs and ongoing care. Because passively acquired cats and kittens bypass the shelter or pound environment, it is much more difficult to provide new owners with information and support regarding sterilization, microchipping, health and wellbeing.

Passive acquisition presents challenges when dealing with cat overpopulation issues. An unfortunate outcome of this, is that whilst residents will often assume caregiving responsibilities for one or more stray cats or kittens, financial constraints or limited access to affordable veterinary or sterilization services can result in ongoing unwanted litters, perpetuating the local overpopulation issue. This highlights the importance of targeted sterilization programs to increase the proportion of cats sterilized.

### 4.3. Cat Containment

Our results indicated that 51% of respondents contained their cat/s completely to their premises, with most being kept completely indoors, and another 18% contained their cat at night. Our results are lower than a recent study where 65% of NSW residents indicated they currently kept their cat(s) fully contained, with a further 24% containing their cats overnight [96]. Of note, in the city of Ipswich cats must be contained to the owner’s property, but this is generally not required in NSW. In comparison, other Australian studies reported lower rates of cat owners keeping their cats exclusively indoors or contained within their property, including 40% [97], 34% [40], and a low of 29% [98]. The more recent Australian containment rates are generally higher than the 37% of UK cat owners [76] who contained their cats, but lower than for US and Canadian cat owners who contained their cats solely indoors (57%) or indoors with controlled access outdoors (22%) either via direct supervision, in an enclosed area, or being kept on a harness or leash [99].

The lower rate of owners containing cat/s all the time in the UK may reflect the cultural acceptance of outdoor cats [100]. In Australia, indoor-only cat policies are strongly advocated, and containment has strong support in the community and appears to be increasing amongst cat owners due to wildlife concerns [101,102]. That over 50% of cat owners in our study fully contained their cats is encouraging, particularly considering the study was conducted in a low socioeconomic area. Barriers in low socioeconomic areas often make cat containment challenging, with rental accommodation in these areas often unsuitable for cat containment due to a lack of cat-proof fencing, inadequate screens on windows or doors and the prohibitive cost of containment systems. These factors would be expected to result in lower containment rates.

Consistent with a previous finding that 14% of surveyed cat owners disagreed with containing their cat, 15% of cat owners in our study did not contain their cat/s at all. In addition, three of the 60 cat owners (5%) who said they ‘always’ contained their cat, indicated that it had escaped or wandered off their property in the previous two weeks. A US study reported that 41% of owners who lost their cat said it was an indoor-only cat [103] highlighting that, even with the best of intentions to contain a cat, they can be difficult to contain. This may explain why 15% of cat owners lose their pet at least once in a 3-year period [104].

An Australian study found that cat owners who were middle-aged and male were more likely to allow their cats to roam [7,96]. In addition, homeowners were less likely to contain their cats all the time compared to renters, apartment dwellers and those with no outdoor space [7,96]. Despite participants experiencing barriers, such as a lack of physical structures or financial impediments required to develop containment structures and provide behavioural enrichment resources, participants indicated a commitment to contain their cats [96,105]. Owners’ beliefs on the importance of containment were due to concerns for wildlife protection, the safety of their cats, and community welfare [40,41,96]. Fear of punitive actions including fines might also be a reason, but this was not assessed in these studies. In a recent Australia-wide study, 71% of cat owners agreed that cats should be confined to their property whenever unsupervised, and 89% agreed that cats should be confined inside the house at night, which is higher than the 69% in our study who contained their cat at night [41]. In contrast, only 54% of dog owners supported containing their dog at night, despite pet dogs being more often responsible than urban cats for predating threatened and endangered wildlife, most of which are nocturnal, with most dog attacks occurring on the owners’ property [41].

Given challenges in disadvantaged areas to containing cats and, in Australia, concerns about wildlife predation, assistance could be offered to low-income communities to install effective enclosures, screens on windows and doors, and air-conditioning, especially where threatened and endangered species are present. In addition, information on the benefits of bedtime feeding [106] could assist containment at night of “door-dasher” cats and protect wildlife of conservation value, because most threatened and endangered species at risk of cat predation are nocturnal [41,107].

### 4.4. Feeding Unowned Animals

Understanding the phenomenon of cat semi-ownership is crucial for several reasons. It has significant implications for animal welfare, public health, and ecological balance. High rates of semi-ownership can complicate efforts to manage stray cat populations, so understanding what maintains these can help inform the development of more effective strategies for controlling stray cat populations over time, improving cat welfare and minimizing cat-related issues in the community. Cat semi-owners have an opportunity to facilitate sterilization of cats they are feeding and take ownership of them [9,20]. This is exemplified by the city of Banyule’s community engagement program that provided free sterilization, microchipping and registration for the cats adopted by their semi-owners, as well as for owned cats in target suburbs with high cat-related calls. Cat-related complaints were reduced by 51% in 3 target suburbs, and cat impoundments and euthanasia city-wide decreased by 66% and 82%, respectively, over 8 years [20]. Savings in cat management costs for the local government resulting from this program amounted to AUD 440,660 [20].

Our results highlight the importance of how questions regarding feeding cats are asked, because 9% of respondents reported feeding cats they did not own, but, when asked whether they knew who owned the cat, only 3–4% of respondents fed a cat that was presumed unowned and therefore would be considered semi-owners. How the question is asked, including specifying daily or regular feeding, likely accounts for some of the substantially higher results reported in the literature. Distinguishing between cats that are known to be owned or stray is important because of the opportunity for interventions to support people feeding stray cats to take ownership of them, for example by providing free sterilization and microchipping [20].

A previous Australian study reported that 3% of adults fed an unowned cat, where socioeconomic status was balanced across a city of 1.2 million [8]. However, other Australian studies reported up to 10% of adults fed unowned or stray cats, and that they fed an average of 1.5 cats [9]. One US study reported 26% of respondents fed a stray cat in the last year, but respondents were not required to specify whether feeding was infrequent over the previous year or was daily or regular feeding. A recent study of seven communities in the US found between 15% and 47% of respondents said they ‘put out food for stray cats in the neighbourhood that do not sleep in their house’ and they fed an average of 2.5 cats [108]. Higher levels of feeding stray cats are reported from some European countries, with 30% of residents in Bulgaria [63], and 17% in Italy indicating they fed stray cats [109] which potentially reflects greater acceptance of stray or street cats and government support for TNR programs.

Given that most semi-owned cats are not sterilized, the potential for them to contribute to the stray cat population is high. For example, even if 3% of adults feed an average of 1.5 cats, in the 3 target suburbs (nearly 40,000 residents; 80% adults), 960 adults would be expected to be feeding 1440 stray cats (36 cats/1000 adult residents or 28 cats/1000 residents), most of which are not sterilized. This is consistent with estimates of 29 free-roaming cats/1000 residents in highly disturbed environments in Australia [3]. Of these 1440 cats, approximately 50% would be expected to be female and 90% unsterilized. They could produce an average of 5 kittens per year (3240 kittens), although 75% would likely die mostly from infectious disease or trauma before 6 months of age [90]. Even if only 25% survive, approximately 810 kittens would be added to the population annually. In comparison, the average pre-COVID intake into the receiving shelters and pounds from these three suburbs was approximately 500 cats/year (unpublished data). Using the same calculation for the 35% of owned cats in our study, of which 9% were unsterilized, these cats would contribute another 3780 kittens, of which about 945 would survive to 6 months. This highlights the need for targeting both semi-owned and owned cats with sterilization programs in areas with high shelter and pound intakes, or high numbers of cat-related calls to council.

For people caring for semi-owned cats, where they are unable to take ownership, for example due to rental restrictions relating to pet ownership and mandated containment laws, it is recommended that the legislation is changed to allow these cats to be sterilized to stop them contributing to the stray cat population. Cat caregivers’ bond with the cats they are caring for is virtually identical to that of cat owners with their owned cats [110] and lethal management of these cats results in severe negative mental health impacts, including symptoms consistent with post-traumatic stress [1]. Therefore, an assistive One Welfare approach to stopping kittens being born is recommended and needs to be facilitated by legislative change.

In our study, a greater proportion of women than men fed unowned cats. This is consistent with previous research [7]. Feeding unowned animals is attributed to a combination of cultural and social factors. Studies have shown that societal expectations often position women as caregivers, influencing their propensity to nurture animals in need [111]. Additionally, women tend to have higher levels of empathy towards animals, further driving their engagement in these activities [7]. Surprisingly, 8% of respondents fed dogs they did not own, which was an unexpected result given that free-roaming community dogs are not a feature of Australian urban areas, in contrast to cats. It is unfortunate that we did not ask dog feeders if they knew if the dog was owned, because it would be assumed most were known to be owned.

### 4.5. Limitations of Study

This study’s design has several limitations that should be acknowledged. First, the high degree of self-selection and use of self-reported data may both have introduced bias, as respondents’ pet ownership and management views may have differed systematically from eligible non-respondents, and responses may have been influenced by respondents’ perceptions and willingness to provide socially desirable answers [112]. In relation to precision, a sample size of over 300 is generally considered sufficient for most quantitative analyses [113]. We used data from 343 respondents for pooled analyses at respondent level but precision was lower for results from subsets of respondents, and for results for their cats (n = 201) and dogs (n = 261). Detailed income data, a known predictor of pet ownership and sterilization practices, was not collected, further limiting full understanding of socioeconomic factors influencing stray cat populations. The COVID-19 pandemic also necessitated a shift from face-to-face to telephone surveys, potentially affecting the response rate and the depth of interaction with participants [114].

The use of only three suburbs in Ipswich, Queensland, limits the generalizability of the findings to other regions with different socioeconomic and demographic profiles as the study suburbs were selected based on high shelter and impound admissions. This was purposeful as we aimed to obtain data from residents in suburbs to be targeted for the Community Cat Program to provide baseline data on sterilization rates, unplanned litters and cat containment. However, the results provide valuable comparative information for other locations undertaking a situational analysis prior to implementing a sterilization program in an area with high cat intake or cat-related calls.

## 5. Conclusions

Current strategies for managing urban stray cats in Australia have proven ineffective, relying on outdated and reactive methods that fail to achieve long-term population control. Whilst sterilization rates for owned cats appear to be high in the targeted areas, they are insufficient to prevent unwanted litters from owned cats contributing to the stray cat population. Cat containment remains a significant challenge for cat owners, further contributing to unwanted litters. In addition, there is a considerable proportion of stray or semi-owned cats being cared for by people who do not perceive them as their own, and these are likely a major contributor to maintaining free-roaming urban cat populations. Facilitating access to a free sterilization and microchipping program would be expected to have substantial benefits for animal welfare, public health, and ecological balance. Further, such a program could reduce government expenditure on animal management strategies that are ineffective. Proactive, assistive approaches to cat management that move away from traditional lethal management methods will reduce the negative mental health impacts on cat carers, veterinarians and shelter staff alike, and improve job satisfaction of animal management officers, and are aligned with a One Welfare philosophy.

Moving forward, effective management strategies for stray cat populations should prioritize targeted, high-intensity sterilization programs for owned cats and include community engagement to transform semi-owned cats into fully owned pets. Legislative changes that allow sterilization and microchipping of cats that remain semi-owned or unowned will be required to maximize the effectiveness of these programs. Addressing the root causes of stray cat populations through proactive measures that increase sterilization rates in owned, semi-owned and unowned cat populations, rather than reactive approaches, enable communities to mitigate the challenges associated with stray cats while improving outcomes for animal and human welfare.

Other legislative changes include waiving registration (licensing) and permit fees for excess cats or those not sterilized by 4 months of age (unless purposely kept for breeding), and replacing mandated containment laws with anti-nuisance laws. It is recommended that where caregivers are unable to take ownership of cats because of state or local bylaws or rental property limitations, these cats are legally able to be sterilized, and rescue groups, animal welfare organizations or businesses be registered as secondary contacts on microchip databases. This is especially important in multi-cat situations where the caregiver cannot take ownership of the cats. If left unsterilized, these cats contribute to stray cat populations, but traditional lethal management causes severe negative mental health impacts on many carers and shelter staff.

Future research should continue to monitor the effectiveness of these strategies and explore innovative approaches to enhance the coexistence between pets and wildlife in urban and peri-urban environments, ultimately fostering sustainable and harmonious relationships between humans and animals which also benefit their environments, supporting One Welfare principles.

## Figures and Tables

**Table 1 animals-14-02849-t001:** Sources from which respondents obtained their owned cats or owned dogs.

Animal Source	% (Number)	% (Number)
	Owned Cats (n = 201)	Owned Dogs (n = 261)
Family friend (including 1 cat “from a client”)	31% (62)	24% (62)
Shelter or council pound	20% (40)	19% (49)
Found (cats: including 1 “appeared as stray”, 1 “saved from crack house”, 1 “dumped”)	19% (37)	0.4% (1)
Online (including 1 (cat) from newspaper)	12% (23)	6% (15)
Pet shop	3% (5)	6% (16)
Registered ^1^ breeder (including 2 from cattery)	8% (16)	23% (60)
Unregistered breeder	4% (7)	13% (34)
Veterinary clinic	2% (4)	1% (3)
Bred ourselves	2% (3)	4% (10)
UQ Gatton Cat therapy program	1% (1)	n/a ^2^
Other	1% (2)	-
Adopted (2 ex-military, 2 retired greyhounds, 1 adopted from Shane Warner Foundation, 1 rescued from shooting, 2 adopted from unspecified source)	n/a	3% (8)
Source not specified	1	3

^1^ Registered breeder is registered by Dogs Australia or affiliated state organization. ^2^ n/a = not applicable

**Table 2 animals-14-02849-t002:** Ages and sterilization statuses of owned cats (n = 201) and dogs (n = 261).

Age Category	Age Range	Cats		Dogs	
		% of All Cats (Number)	% (n) of Age Category That Were Sterilized	% of All Dogs (Number)	% (n) of Age Category That Were Sterilized
Kitten/puppy	<4 months	4% (7)	29% (2)	2% (4)	0% (0)
Juvenile	4–6 months	5% (9)	44% (4)	4% (10)	40% (4)
Young adult	7–11 months	5% (10)	100% (10)	3% (8)	50% (4)
Adult	1–6 years	50% (99)	95% (91) ^1^	52% (135)	74% (100)
Senior	7–15 years	34% (67)	97% (65)	38% (99)	79% (78)
Geriatric	>15 years	4% (8)	100% (8)	2% (4)	100% (4)
Not recorded			100% (1)	^2^	^2^

^1^ 95% (91/96) as sterilization status was not recorded or not known for 3 adult cats. ^2^ For one dog, neither age category nor sterilization status were recorded.

**Table 3 animals-14-02849-t003:** Cat containment, methods used to contain cats all the time, and timing of partial containment described by 117 cat owners.

Degree of Containment; Methods Used for Cats Fully Contained; and Timing of Partial Containment	(%, n/60 or n/39)
*Contained all the time: 51% (95% CI: 42% to 60%; 60/117)*	
Kept indoors entirely	68% (41/60)
Kept indoors, with access to an outdoor enclosure	13% (8/60)
Indoors at night, yard during the day which has fencing additions to prevent cats getting out (e.g., inverted/rollers/electric)	8% (5/60)
Day and night cat/s have access to yard which has fencing additions to prevent cats getting out (e.g., inverted/rollers/electric)	8% (5/60)
Other (“Lived outside”)	2% (1/60)
*Contained some of the time: 33% (95% CI: 25% to 42%; 39/117)*	
Always at night	38% (15/39)
Sometimes at night	15% (6/39)
Sometimes during the day	15% (6/39)
Always at night and sometimes during the day	15% (6/39)
Always during the day	10% (4/39)
Other or time of day when contained not specified	5% (2/39)
*Not contained: 15% (95% CI 10% to 23%; 18/117)*	

## Data Availability

Most relevant data are reproduced in the text.

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
