# Peer review of "Situational Analysis of Cat Ownership and Cat Caring Behaviors in a Community with High Shelter Admissions of Cats"

_animals, 2024, doi:10.3390/ani14192849_

Round 1

Reviewer 1 Report

Comments and Suggestions for Authors

Comments on the manuscript “Situational analysis of cat ownership, semi-ownership and cat caring behaviors in a disadvantaged Australian community.” submitted to Animals

Knowing a pet population and a region precedes the action of management plans, castration, reproduction, and assistance to animal welfare. Public health and wildlife preservation programs also require knowledge about the pet population and human-animal relationships.

The present study is a “situational analysis” of domestic cats and dogs living in three city districts in Australia. Using a questionnaire administered in person or online, due to the restrictions of the COVID-19 pandemic, the authors found that most domestic cats in Ipswich, Queensland, have owners, live in a closed indoor system, and are neutered. However, a small proportion of cats are fed but do not have owners. The percentage of young cats that are not sterilized is worrying because they can reproduce without control, with unwanted offspring. Cats are usually acquired from friends, relatives or found. Dogs are also acquired from friends, but many are acquired from breeders. Compared to cats, dogs are preferred as pets by the Ipswich community.

Compared to other locations in Australia, cats in the Ipswich region are under control and appear promising for neutering and adoption programs. Concerning other countries, the situation of cats in Ipswich follows desirable standards of control and welfare.

The study compares cats and dogs in the region. However, the present study does not compare dogs with dogs studied in other regions of Australia and the world.

At the conclusion of the study, the authors argue that it is necessary to carry out interventions to minimize the impact of ownerless cats on the availability of animals for shelters, which will later be euthanized if they are not adopted.

The study is relevant because raising cats within sanitary and animal welfare standards that are ethically and legally acceptable is important. The originality lies in a study carried out in a medium-sized city with a population apparently less wealthy than other places. Scientific articles addressing the relationship between pets and people living in less advantageous neighborhoods are rarely published. However, it is necessary to rewrite and inform some parts of the text that are confusing or omitted.

In general, five parts of the text are weak;

1. First, there is a detailed methodology for understanding whether the data has any sampling bias.

2. The second is the redundancy in writing some data and the weak explanation of what was found in the results.

3. Third, the connection of data about dogs in a cat-focused study (see the title, for example).

4. The lack of explanatory depth of some of the results.

5. Excessive references, many of which are redundant and unnecessary.

Below, I comment in more detail on some of these points.

Title

Line 2: The research included a comparison between cats and dogs. Since the title does not mention the inclusion of dogs in the research, I suggest modifying it.

Simple Summary

It is good.

Abstract

It is good.

Keywords

Line 47: It does not appear that the study has the concept “One Welfare” as one of the main topics. Only twice (lines 504 and 547) does the term One Welfare appear. The text does not address the well-being of humans and their pets. The term “One Welfare” does not seem essential for indexing.

Introduction

This section is good.

Materials and Methods

Line 151: How were the respondents recruited? Was there randomness in the sampling of respondents? Could the sample have any sampling bias? What sampling bias?

How was the sample size estimated?

Results

I did not find the exact p-values calculated using Fisher’s tests (see methods).

I suggest combining Tables 1 and 2 into one table to compare the origins of cats and dogs better.

The results are redundant because they appear in the text and in tables A2, A3, and A4. Some of these values are unnecessarily restated in the discussion (see below). In the technique of scientific article writing, this way of presenting results is technically counterproductive and boring for the reader. I suggest representing the results only in tables instead of writing each result verbatim again in the body of the text.

Line 235: What about the containment of dogs? Why were no results presented in this subsection?

Discussion

The weakest part of the manuscript is the Discussion. I point out two big problems:

1. Repetitive and redundant writing of result values (e.g. lines 297, 310, 212-215, 390-391, 395, 454, 458-459 etc.) In the Discussion, the values must be interpreted but not shown again. As it stands, it is a shallow and boring interpretation. The author must deepen, qualify, contextualize, and provide interpretative contours for each value in the results section.

2. There is a lack of critical argumentation of its results, comparing them on theoretical and empirical bases with other works. It is not enough to cite results from other articles, stating that the results had similar or different values (higher or lower) from those found in the study (e.g., lines 389-401; 454-467 etc.). The author should go further and try to explain the causes and consequences of the findings as he did in the good discussion on the topic “Sterilization and unwanted litters” (lines 333-387).

I suggest rewriting the discussion to consider these critical, interpretative, and relevant aspects in order to understand the results.

Line 438: The attitude of containing your dogs and cats is attributed to “Owners’ concern for wildlife protection, the safety of their cats and community welfare”. Couldn't the owners' attitude also be linked to the fear of being prosecuted by the courts and receiving fines?

Author Response

Thank you very much for your valued input into our paper, which has resulted in it being a more valuable contribution to the literature. Please see the attached file with detailed responses.

Reviewer 2 Report

Comments and Suggestions for Authors

Rand…check order of first and last name.

This is an interesting paper about cat and dog ownership, pet keeping, and feeding of animals not considered to be owned in 3 disadvantaged suburbs. The purpose was to provide a baseline and information to inform a free cat sterilization problem. Generally, the manuscript need to be more focused on the objectives and to better connect the data to the ultimate purpose. Some questions to consider in focusing the manuscript (particularly the introduction and discussion) are 1. Why is the comparison between dogs and cats important? And if it is, articulate that clearly and present the data in the results section accordingly; 2. What are the results important for these suburbs and how will they inform the intervention? And 3. Which of these baseline data are expected to change with the intervention and are therefore most important for this study and manuscript? I also have additional specific comments below.

Line 23: in many countries, finding or taking in a stray cat is quite different and more frequent than getting from a friend. Was this the situation here? And would be helpful for comparison to break this, perhaps just for the abstract—I saw the data in the manuscript. (Same for line 36.)

Line 26: on line 253, the data state 9% (which is much more similar to other studies in other countries). I don’t believe that the estimate for which cats are owned is as important as the overall one but if the authors feel both are important, they should be clearly differentiated throughout the manuscript and the importance emphasized.

Line 38: this sentence isn’t clear. I think that only 62% of cats under 1 year old were sterilized? Please clarify in text. And there were data on cats older than 3-4 months to 1 year which seems like it would be even more helpful!

Line 56: please reference this definition as it does vary.

Line 64: no, these are not “wild” cats in the US. Depends on the reference but feral means many things in different publications. Usually in community cat work, it means unsocialized or unaccustomed to humans and unowned.

Paragraph starting Line 68: please clarify what country(s) is meant here. Given the several Australian references I think this could be relative to this country. And then the other references could be included as comparison or confirmation. This is important because “feral” cats as defined in some countries DO enter the shelter system. The rest of this paragraph is confusing; please use one set of terms throughout the manuscript. Please edit.

Line 102: move to start of next paragraph and rephrase to indicate that containment has not been shown to be an effective strategy and reference if that is the case.

Line 108-10: please indicate where this is happening and reference if possible. Otherwise, the following sentence doesn’t make sense.

Lines 152-58: How were respondents within households selected (apart from being over 18) for both in person and telephone? Is the 3121 the number of phone numbers? If so, how were the phone numbers selected for calls? How many times called, knocked? The methodology needs to be better explained so that the potential biases in the survey results can be better identified.

Line 156-63: these are results I think as the survey and response rate are important elements to interpret response bias. Same for lines 168-70.

Table 2: if the main comparison is between dogs and cats, it would make sense to combine table 1 and 2, with the order as written from the cat table. 

Table 3: always at night and sometimes during the day is missing a / between 6 and 39.

I’d like to see 95% confidence intervals for the main outcome variables (for example, contained all the time, contained some of the time, not contained—perhaps for the key baseline measurements) to provide some measure of precision with this sample size. This is particularly important as no justification for sample size was given.

I don’t see any statistical analysis although it is listed in the methods and should be performed as the manuscript is currently written. Please specify which variables were intended to be compared in the methods, and then what p-values were found in the results.

Discussion:

The discussion is a bit wordy and long, often repeating results several times without tying those findings to the purpose of this survey: to plan for an intervention and see what baseline is. Are there specific data points in the baseline that the authors would expect to see change with this intervention? Using that lens could help focus the discussion better. Paragraph starting on line 446 is more what I think could be helpful.

Why are the results of the other species important (except where dogs and cats are important to compare—which the reader won’t know about till the statistical analysis section is updated)? This manuscript and intervention is about cats. Please consider and revise discussion accordingly.

Line 284-5: please reference this statement (and if possible, the one before) even if grey literature is the only source.

First paragraph: also note the low frequency of earlier age sterilization for cats and if that is readily available in this part of Australia (given that they can become pregnant as early as 5 months and that all litters were accidental). These are key findings for targeting human behavior change.

Lines 302-5 should be in the methods as how the survey is framed can be crucial. Also, how the person who actually completed the survey was selected is important. If it was whoever answered the door or phone, that would make sense given lines 301-2.

Lines 305-9: This pattern is seen in every study I’ve ever read. Could be due to caring but also is a consistent finding about who identifies as the pet owner.

Line 313-4: Please also remind the reader here that SEIFA scores were used for the suburbs which is why (I’m assuming) that income wasn’t included? And how would these scores relate to income or education? Explaining more about the content of these scores (perhaps including a bit more about how these locations were chosen in methods) could be useful context here.

Section 4.2: 95% confidence intervals for the present study and if possible, from the referenced studies would allow conclusions to be drawn about whether these differences in pet ownership are statistically different or not.  Similar comment for dog vs cat.

Section 4.2.1: the authors didn’t ask if any of the cats had a litter prior to sterilization. That is often a yes answer even with very high sterilization rates! Something to consider in the future and an addition to the manuscript please. Line 350: and intact male cat urine stinks, and female cats in heat make LOTS of noise so much more motivation to address. Please add to the section.

Paragraph starting on line 369: If the present study doesn’t match the assumptions for that model, the data for the conclusions aren’t valid. Please edit accordingly.

4.2.2: additionally, it is very difficult to connect to new cat owners and provide them with information or resources when they get cats passively; when cats are acquired from shelters or other intentional sources, the cats can more often already have critical veterinary care and the owners can be engaged in conversations about cat needs. Please add something about this to this section, it isn’t fully explained here.

Reference 103 is incomplete. And I think that depending on the reference indoor only cats in the US are even higher than this.

Line 432 and following: I can’t tell what is from the reference, what country or socioeconomic status are included and what is about the current study. Please clarify in the text.

Line 455: This information needs to be included in the results. And the definitions need to be in the methods. Also, it isn’t clear why this matters especially if the respondent wasn’t really sure the cat wasn’t owned. Please edit.

Line 463: where are these other country’s data from? Please include primary source references here if these are not primary sources.

Line 465-6: and why is this important for the present study? Feels out of place here but could be important overall.

Line 476-83: why is this important for the present study? And how does it relate to the 3 suburbs?

Line 493: reference?

Limitations: please add limitations from the sample size and selection of participant.

Author Response

(The authors gave the same response as above.)

Round 2

Reviewer 1 Report

Comments and Suggestions for Authors

Comments on the manuscript “Situational analysis of cat ownership and cat caring behaviors in a community with high shelter admissions of cats” submitted to Animals

 The manuscript has been substantially modified, now presenting more detail without unnecessary redundancies and prolixity. In this text, the methods have been explained in detail, dispelling doubts about the study design and any bias. There is no doubt that the scientific rigor of the analysis has been achieved. Notably, the discussion has improved, with rich reasoning about the importance of bringing together actions and many social actors to control and manage the cat population in Australia. Based on this study's observations, the proposed strategy converges solidly with the principles of One Welfare. 

This article is promising and will be influential in supporting many studies on the management and control of the cat population in Australia and other countries.

 I congratulate the authors for presenting an article that is now excellent for the multidisciplinary field of knowledge that encompasses population management, public health, conservation, and the welfare of cats and urban communities.

Author Response

Thank you so much for your supportive comments!

Reviewer 2 Report

Comments and Suggestions for Authors

The authors have really improved the clarity and completeness of the manuscript! It is a much stronger and more applied paper now.

I might suggest that table A2 be divided into cat and dog owners. These demographics by this breakdown are not otherwise in the manuscript.

Line 501: being received as a gift was not associated with any challenges in the US. And people can passively acquire pets when someone passes away or is imprisoned. That is fairly common in the US at least. Please edit.

Author Response

Once again, thank you very much for your constructive comments and insights during the peer review process. Your ongoing feedback has been invaluable in enhancing the quality and clarity of our manuscript.

Please find the responses below and the corresponding revisions in the manuscript.

  1. I might suggest that table A2 be divided into cat and dog owners. These demographics by this breakdown are not otherwise in the manuscript.

Response: Thank you for your suggestion regarding this breakdown. The breakdown for gender versus dog and cat ownership is in the results and referred to in the discussion. Although investigating dog and cat ownership by age and education would be interesting, we feel that it is outside the scope of our study and would be best investigated in a study specifically designed to look at these factors, for example adding in income and housing type amongst other items for investigation. Thus we elected to not modify the information in Table A2. For readers interested in this information, these details are cited in the text in Results 3.1 and 3.2. We have included them below for you information.

3.1. Cat ownership - Of the 332 respondents who recorded their gender, 38% (85/221) of female respondents were cat owners, compared to 29% (32/111) of male respondents.

3.2. Dog ownership - Of the 332 respondents who recorded their gender, 57% (126/221) of female respondents were dog owners, compared to 45% (50/111) of male respondents.

  1. Line 501: being received as a gift was not associated with any challenges in the US. And people can passively acquire pets when someone passes away or is imprisoned. That is fairly common in the US at least. Please edit.

Response: Thank you for suggesting this addition, we have modified the text to include:

” Passive acquisition occurs when cats are unintentionally obtained, including being found or obtained through social networks or received as gifts from family or friends [7]. This includes acquisition when someone in their network is no longer able to care for their cats due to life changes, (eg entering nursing home, passing away, imprisonment, relocating overseas or extended travel) [92]. There was no difference in attachment between pets who were, or were not, acquired intentionally, and one study reported that pets acquired unintentionally as gifts or free were less likely to be relinquished than pets acquired intentionally [92,93].

Thank you once again for your invaluable contribution to our work.
